# Development of Multiplex PCR Assay for Screening of T6SS-5 Gene Cluster: The *Burkholderia pseudomallei* Virulence Factor

**DOI:** 10.3390/diagnostics12030562

**Published:** 2022-02-23

**Authors:** Noreafifah Semail, Azian Harun, Ismail Aziah, Nik Mohd Noor Nik Zuraina, Zakuan Zainy Deris

**Affiliations:** 1School of Medical Sciences, Universiti Sains Malaysia, Kubang Kerian 16150, Kelantan, Malaysia; noreafifah93@gmail.com (N.S.); azian@usm.my (A.H.); 2Institute for Research in Molecular Medicine (INFORMM), Universiti Sains Malaysia, Kubang Kerian 16150, Kelantan, Malaysia; aziahismail@usm.my

**Keywords:** *Burkholderia pseudomallei*, T6SS-5, optimization, multiplex PCR

## Abstract

Despite the advanced understanding of the disease, melioidosis, an infection caused by *Burkholderia pseudomallei*, continues to be of global interest. The bacterial virulence factor, type six secretion system-5 (T6SS-5), in particular, is an essential factor for *B. pseudomallei* that is associated with internalization and intracellular survival of the pathogen. To detect the virulence gene cluster, this study has successfully developed a novel seven-gene (*tss*C-5, *tag*D-5, *tss*A-5, *hcp*-5, *tss*B-5, *tss*F-5, and *vgr*G-5) multiplex PCR assay. The optimum annealing temperature for this assay ranged between 59 and 62 °C. The limit of detection for this assay was 10^3^ CFU/mL for all genes, excluding *tss*F-5, which was found at 10^5^ CFU/mL of the bacterial concentration. In sensitivity and specificity tests, this multiplex assay was able to amplify all of the seven target genes from 93.8% (*n* = 33/35) clinical and 100% (*n* = 2/2) environmental isolates of *B. pseudomallei*. Whereas only four genes (*tss*C-5, *tag*D-5, *tss*F-5, and *vgr*G-5) were amplified from *Bukholderia thailandesis*, two genes (*tag*D-5 and *tss*B-5) were amplified from *Bukholderia stagnalis,* and zero target genes were amplified from *Bukholderia ubonensis*. No amplification of any genes was obtained when tested against isolated DNA from non-*Bukholderia* species (*n* = 20), which include *Staphylococcus aureus*, *Klebsiella pneumoniae*, *Enterococcus faecalis*, and others. In conclusion, this multiplex PCR assay is sensitive, species-specific, rapid, and reliable to detect the virulent gene cluster T6SS-5 of *B. pseudomallei*.

## 1. Introduction

*Burkholderia pseudomallei* is a gram-negative environmental saprophyte with high genetic diversity [1]. It is the causative agent of melioidosis, a neglected tropical and potentially fatal disease affecting both humans and animals. Melioidosis, a nominated Tier 1 select agent by the U.S. Centers for Disease Control and Prevention (CDC), is endemic in Southeast Asia and Northern Australia. It has expanded, and has been reported worldwide [2,3]. Every year, global melioidosis in humans is estimated at around 165,000 cases, with a mortality rate of over 50%, a burden which is similar to that of measles [4,5]. In Malaysia, it is estimated that more than 2000 patients die due to melioidosis per year, which is much higher than the death caused by tuberculosis or dengue fever based on the reported incidence and mortality rate [6]. This indicates that the pathogen is highly virulent.

Type six secretion system-5 (T6SS-5) is one of the major virulence determinants for *B. pseudomallei* that has a central role in bacterial intracellular life cycle in mammalian host cells [7,8]. T6SS-5 is also crucial for the pathogenesis of systemic melioidosis [7,9,10,11]. T6SS-5 has been constantly revealed as an important component for intercellular spread, multinucleated giant cell formation (MNGC), and virulence mechanisms. On the other hand, other T6SS clusters were reported to have different functions, especially in the persistence of this bacteria in its environmental niches. For instance, T6SS-1 and T6SS-4 are involved in inter-bacterial species competition and the acquisition of metal ions, respectively [7,10,12].

The T6SS-5 cluster comprises 13 genes and 2 accessory subunits that encode for the core components in the accumulation of a functional T6SS, and for the tags or regulator genes, respectively [12]. Although the function of these accessory subunits, four tag genes (*tag*A/B-5, *tag*B-5, *tag*C-5, and *tag*D-5) and two *vir*AG regulator genes, is currently unknown, they are required for transcriptional activation of T6SS-5, and for the accurate assembly of the secretion apparatus during infection [13]. Of note, T6SS was induced throughout the melioidosis infection, highlighting its significant role to ensure *B. pseudomallei* persistence and replication in the cell cytosol [4]. Therefore, the determination of genetic variation in the virulence factors, particularly the T6SS-5 gene cluster, is vital towards a better understanding of its correlation with the clinical presentations, relapse, and outcomes of melioidosis.

Research on *B. pseudomallei* has become of great interest, owing to the bio-threat potential of this pathogen, and an increasing awareness of melioidosis and its burden [14]. For that reason, the purpose of the present study was to develop a sensitive, species-specific, rapid, and reliable multiplex PCR assay for the detection of seven genes of the T6SS-5 virulence gene cluster. This assay might be useful for the early screening of potential virulence strains of *B. pseudomallei* from both clinical and environmental sources. In addition, this assay could facilitate in the laboratory diagnosis of melioidosis, since the pathogen is often misidentified with its closely related species.

## 2. Materials and Methods

### 2.1. Bacterial Strains and Culture Conditions

Archived *B. pseudomallei* clinical isolates and other reference bacterial strains used for the development of multiplex PCR in this study were obtained from the Department of Medical Microbiology and Parasitology, Hospital Universiti Sains Malaysia (HUSM). The environmental isolates were obtained from the Institute for Research in Molecular Medicine (INFORMM), USM, which include *B. pesudomallei* isolates and other *Burkholderia* species, namely *Bukholderia thailandesis*, *Burkholderia ubonensis*, and *Burkholderia stagnalis*. The clinical and environmental isolates were initially validated via a Vitek-2 automated instrumentation system. The isolates were sub-cultured onto a Mueller–Hinton (MH) agar plate, and incubated at 37 °C for 48 h. The pure colonies were used for further DNA isolation. For long-term storage, all isolates were stored at −80 °C in Brain Heart Infusion (BHI) broth containing 20% glycerol.

### 2.2. Number of Colony Forming Units (CFUs) of Burkholderia pseudomallei

*B. pseudomallei* was sub-cultured onto MH agar plates, and grown overnight at 37 °C. A single colony was collected and suspended in 5 mL MH broth. The culture was incubated at 37 °C until the optical density reading at 600 nm (OD 600) reached 1.0. Ten-fold serial dilution was performed from the culture, by which 100 µL and 1 mL of each dilution factor was plated onto MH agar plates, and underwent DNA extraction by using a boiling method, respectively. The CFU of *B. pseudomallei* was counted following 24 h of incubation.

### 2.3. Specific Primer Design

All primers for the T6SS-5 gene cluster were designed using the National Center for Biotechnology Information (NCBI) Primer-BLAST, based on the available *B. pseudomallei* genome information (GenBank accession number: CP002834). In silico specificity of the primers was initially analyzed using the NCBI standard nucleotide-BLAST. The primers used in this study were synthesized by the Integrated DNA Technologies (IDT, Singapore). Primer sequences and their amplicon sizes are shown in Table 1.

### 2.4. DNA Template Preparation

A bacterial DNA template was prepared by using a boiling method, where the bacterial colony was mixed with 200 µL of distilled water by pipetting up and down several times. The mixture was boiled at 100 °C for 10 min, and was centrifuged at 13,000× *g* for 5 min. The supernatant was collected into a clean micro-centrifuge tube, and stored at −20 °C.

### 2.5. PCR Amplification

For the development of multiplex PCR, each 20 µL reaction contained 5 µL of MyTaq Red reaction buffer, and 0.3 µL of MyTaq Red DNA polymerase (Bioline Reagents Ltd., London, UK), 4.7 µL of dH_2_O, 0.5 µL of each primer pair, and 3 µL of DNA template. A PCR run was performed in a thermal cycler machine (Eppendorf, Hamburg, Germany). The PCR cycling condition was set up with an initial heating step at 95 °C for 5 min, plus 30 repeating cycles consisting of denaturation at 95 °C for 30 s, annealing at 61 °C for 30 s, and elongation at 72 °C for 1 min. A final elongation step at 72 °C for 5 min was added to complete the polymerization process.

### 2.6. Agarose Gel Electrophoresis

After PCR amplification, the PCR products were analyzed using 2% agarose gel in 0.5 × TBE electrophoresis buffer. Gel electrophoresis was run at 80 volts for 90 min to separate the target bands. The DNA fragment in agarose gel was visualized and photographed under the ultraviolet light of a G-Box image analyzer (Syngene, Frederick, MD, USA).

### 2.7. Optimization of Multiplex PCR

The multiplex PCR was optimized to ensure appropriate amplification, and to obtain the greatest intensity of the PCR bands. The primers concentration was optimized by reducing the concentration of *tss*A-5, *tss*C-5, and *tag*D-5 from 10 µm of the initial concentration to 5.0, 2.5, 1.25, and 0.625 µm. The annealing temperature, ranging from 52 °C to 62 °C, was also optimized using a gradient program on a thermal cycler.

### 2.8. Analytical Sensitivity of the Multiplex PCR Assay

Analytical sensitivity of the multiplex PCR assay was evaluated by using a ten-fold serial dilution of *B. pseudomallei* overnight culture. DNA was prepared from each dilution by using the boiling method described earlier. Three microliters of DNA from each dilution were used as the template for evaluating the limit detection of the multiplex PCR assay. The PCR was run under the conditions described above.

### 2.9. Accuracy Test of the Multiplex PCR Assay

To validate the efficiency of the multiplex PCR assay, extracted DNA from *B. pesudomallei* clinical isolates (*n* = 32) and environmental isolates of *Burkholderia* species (*n* = 11) were subjected to the multiplex PCR assay. The environmental *Burkholderia* species were two *B. pseudomallei,* two *B. thailandesis,* one *B. ubonensis,* and six *Burkholderia stagnalis* isolates. The specificity of the multiplex PCR assays in this study were verified by using different bacterial species, which include *Staphylococcus aureus* ATCC 25923, *Klebsiella pneumoniae* ATCC 70063, *Enterecoccus faecalis* ATCC 29212, *Pseudomonas aeruginosa* ATCC 27853, *Neisseria meningitidis* ATCC 13090, *Vibrio parahaemolyticus* ATCC 17802, *Escherichia coli* ATCC 35218, *Streptococcus* Group A and B, *Staphylococcus epidermidis*, *Salmonella enterica* serovar Typhimurium, *Streptococcus viridans*, *Stenotrophomonas maltophilia*, *Helicobacter pylori*, *Vibrio cholerae*, *Salmonella* sp., and *Proteus mirabilis*.

## 3. Results

### 3.1. Number of Colony Forming Units (CFU) of B. pseudomallei Clinical Isolate

Colony forming units were used to estimate the number of viable cells of *B. pseudomallei* clinical isolate. At the OD600 of 1.0, the plate counting result showed that the *B. pseudomallei* culture was equivalent to 1.62 × 10^11^ CFU/mL.

### 3.2. Amplification of T6SS-5 Genes

All of the designed primers for T6SS-5 genes successfully amplified 103 bp of *tss*C-5 gene, 155 bp of *tag*D-5 gene, 230 bp of *tss*A-5 gene, 331 bp of *hcp*-5 gene, 406 bp of *tss*B-5 gene, 542 bp of *tss*F-5 gene, and 627 bp of *vgr*G-5 gene (Figure 1, lanes 1 to 7, respectively). All of the seven genes amplified using this multiplex PCR assay are shown in Figure 1 (lane 8). The PCR products were divergent to one another according to their discrete sizes produced on the gel electrophoresis.

### 3.3. Optimization of the Multiplex PCR

The optimal primer concentration and annealing temperature of the multiplex PCR assay were determined to ensure an appropriate amplification and uniform intensity of all of the target genes simultaneously. Of the different concentration tested (5.0, 2.5, 1.25, and 0.625 µm), the primer concentration of 5.0 µm was chosen as the optimal concentration for *tss*A-5, *tss*C-5, and *tag*D-5, as shown in Figure 2.

The optimization of the annealing temperature using gradient PCR in Figure 3 shows that an annealing temperature of 52–62 °C was able to amplify all of the targeted genes. In this study, the intensity of all target bands was observed to be clearer at higher sets of temperature (59 °C to 62 °C). Hence, the annealing temperature at 61 °C was chosen to run the multiplex PCR assay.

### 3.4. Specificity of the Multiplex PCR

The specificity of the multiplex PCR assay was evaluated on *B. pseudomallei* isolates and other bacterial strains, which include *S. aureus*, *K. pneumoniae*, *E. faecalis*, *P. aeruginosa*, *V. parahaemolyticus*, *N. meningitidis*, *E. coli*, *Streptococcus* Group A and B, *S. epidermidis*, *S. enterica* serovar thypimurium, and *S. viridans*. As shown in Figure 4, no amplification was obtained from non-*Burkholderia* strains tested with the specific primer pairs in detecting T6SS-5 genes.

### 3.5. Analytic Sensitivity of the Multiplex PCR

In this study, to determine the analytical sensitivity, lysate DNA prepared from 10-fold serial dilutions of *B. pseudomallei* overnight culture (OD600 = 1) were used as the template in the developed multiplex PCR assay. As shown in Figure 5, the detection limit of the multiplex PCR assay was 10^3^ CFU/mL. Most of the T6SS-5 target amplicons can be detected at 10^3^ CFU/mL. However, the lowest detection limit for tssF-5 gene was 10^5^ CFU/mL, which is two-log higher than other targets.

### 3.6. Evaluation of Multiplex PCR Using Clinical and Environmental Isolates

The efficiency of the multiplex PCR assay was evaluated on 32 clinical isolates of *B. pseudomallei* and 11 environmental isolates (two *B. pseudomallei*, and nine other *Burkholderia* species). When tested on clinical isolates, this multiplex assay was able to amplify all of the 7 genes in 30 isolates. However, two isolates (lanes 16 and 26) were found positive for only five and three genes, respectively. This study also found a variation in the size of these genes, especially tssA-5 (Figure 6a,b). The evaluation of this multiplex assay against the environmental isolates indicated similar results, in which all of the target genes are present in both *B. pseudomallei* isolates. However, only four of the target genes (*tss*C-5, *tag*D-5, *tss*F-5, and *vgr*G-5) were detected from both *B. thailandesis*, two genes (*tag*D-5 and *tss*B-5) were detected from two of six *B. stagnalis*, and no single gene was detected from *B. ubonensis* (Figure 7).

## 4. Discussion

In recent years, the development of specific and sensitive molecular methods for the simultaneous detection and differentiation of *Burkholderia* species has been the subject of growing interest [15,16,17]. However, to our knowledge, this present study is the first study on the development of a molecular-based assay targeting the virulence genes from the Type 6 Secretion Systems-5 (T6SS-5) gene cluster in *B. pseudomallei*. This study has successfully developed a multiplex PCR assay for the screening of seven genes of the T6SS-5 gene cluster from pure colonies of both clinical and environmental isolates of *B. pseudomallei* in a single tube reaction. The developed multiplex PCR assay consists of seven pairs of primers, MyTaq Red reaction buffer, MyTaq Red DNA polymerase, and dH_2_O, in which target amplicon ranging from 103–644 bp were developed for the multiplex PCR assay.

Primer design, as well as primer specificity, is the first and most critical step in the process of establishing a successful molecular detection assay. Therefore, to enable for correct amplification of all of the target genes, all of the primer pairs were designed to suit the following conditions: having an equal range of melting temperature (57–60 °C) and GC content (50–60%), minimal formation of dimers or hairpin structures (delta G value more than −9.0 kcal/mole), and the ability to target the conserved sequence regions [17,18]. In this study, the multiplex PCR was developed to target specific genes of the T6SS-5 virulence gene cluster, which include *tss*C-5, *tag*D-5, *tss*A-5, *hcp*-5, *tss*B-5, *tss*F-5, and *vgr*G-5 genes.

Most of these genes are located in the cytoplasmic (inner membrane) of *B. pseudomallei*, and have important roles in the bacterial pathogenesis. The hemolysin co-regulated protein (Hcp) is a hallmark and a critical component of a functional T6SS, as it facilitates translocation of small effector proteins into the host cell membrane by creating tubules [9,11]. Valine-glycine repeat protein (Vgr) is necessary for MNGC formation or cell fusions, and consequently leads to intercellular spread of the bacterial infection [19,20]. It penetrates the target cell through its needle-shaped β-helical domain [20]. In addition, *tss*A, *tss*B, and *tss*C are involved in the assembly of the baseplate and the sheath-tube, contractile sheath polymerization, and intracellular replication and formation of MNGC, respectively.

It is notable that setting up monoplex PCR reactions in detecting different genes of T6SS-5 gene clusters at a single time is tiresome. Using this multiplex PCR assay, detection and screening of targeted genes of the T6SS-5 gene cluster in clinical and environmental isolates of *B. pseudomallei* would be easier and faster. In comparison to a monoplex PCR that requires lengthy procedures for multiple targets, a multiplex PCR assay could minimize the tedious steps of calculation, preparation, optimization, and pipetting, which can reduce the time consumed for the PCR work, and consequently reduce the chances for carry-over contamination [21]. An approximate duration of four hours is needed to obtain the results of the multiplex PCR in one run, starting from the PCR set-up to the final step of the interpretation using gel electrophoresis.

The annealing temperature of the multiplex PCR assay was optimized using a gradient thermal cycler to test the specificity and sensitivity of primer–template binding. The optimization of the gradient annealing temperature in Figure 2 shows that an annealing temperature of 52–62 °C was able to amplify all of the targeted genes, and 61 °C was chosen as the optimal temperature for this multiplex amplification. Besides that, the developed multiplex PCR assay was highly specific. As shown in Figure 4a,b, there was no target gene amplification observed when tested with other bacterial pathogens, except for a band which is complementary to the amplicon size of the tssF-5 gene in *V. cholerae* isolate.

The limit of detection of a multiplex PCR assay depends on the product size and the combination of multiple primers in a single tube, which might affect the effectiveness of the PCR amplifications [22,23]. In most cases, the sensitivity of multiplex PCR assay is often reduced with the increased number of target genes in the assay [24], which is contrary to our result. The limit of detection of this multiplex PCR assay was 1.62 × 10^3^ CFU/mL for most of the targeted genes. However, the analytical sensitivity for the *tss*F-5 gene (1.62 × 10^5^ CFU/mL) was two-log lower than other targets. Most of the target genes in this multiplex PCR assay could be detected at low amounts of bacterial concentration (10^3^ CFU/mL), thus showing that this assay has a comparable sensitivity to other in-house PCR (10^5^ CFU/mL), lateral flow-recombinase polymerase amplification (10^3^ CFU/mL), and TaqMan Real-Time PCR (10^3^ CFU/mL) assays from previous studies [25,26].

Furthermore, the multiplex PCR assay was evaluated using clinical and environmental isolates of *B. pseudomallei* and other *Burkholderia* species, as shown in Figure 6a,b and Figure 7. Based on the result, of the total 35 *B. pseudomallei* clinical isolates, this multiplex PCR was able to detect all of the target genes from 33 isolates. Interestingly, this study has observed that there was a variation in the size of the *tss*A-5 gene that will be further analyzed using a DNA sequencing method. For the environmental isolates, both isolates of *B. pseudomallei* can detect all of the seven T6SS-5 targeted genes. However, only four of the target genes, which include *tss*C-5, *tag*D-5, *tss*F-5, and *vgr*G-5, can be detected from *B. thailandesis*. This assay could not detect any of the target genes from *B. ubonensis*. However, *tag*D-5 and *tss*B-5 genes can be detected from two isolates of *B. stagnalis*. This result indicates that not all genes in the T6SS-5 gene cluster are present in all *Burkholderia* species, which might be due to the lesser pathogenicity of other species. One limitation of this study is that the sensitivity, specificity, and evaluation of the clinical and environmental isolates were tested on a limited number of isolates. An inadequate number of *B. pseudomallei* environmental isolates was available from our collection. In future accuracy tests, the sample size and type of organism should be expanded, and diagnostic evaluation of this assay directly from clinical specimens should be included for an extensive screening. Other *Burkholderia* species, such as *Burkholderia mallei*, *Burkholderia cepacia*, and *Burkholderia vietnamiensis*, and non-Burkholderia species, particularly *Chromobacterium violaceum*, which can cause an infection with similar symptoms to melioidosis, should also be included for a more comprehensive analysis. Apart from that, further studies are also needed to develop an assay consisting of the whole set of genes in the T6SS-5 gene cluster to elaborate the function of this particular virulence factor in *B. pseudomallei.*

## 5. Conclusions

In conclusion, this study has successfully developed a novel, sensitive, species-specific, rapid, and reliable multiplex PCR assay for the detection of the T6SS-5 virulence gene cluster of *B. pseudomallei* from pure colonies. This assay is convenient and time-saving, which is applicable in many places compared to other expensive techniques, such as real- time PCR, next generation sequencing, and micro-array, which have been used to identify this bacterium. It is also relevant for the screening of *B. pseudomallei* strains from both clinical and environmental sources. Moreover, the developed PCR assay can be used to identify and distinguish *B. pseudomallei* species from other bacterial strains by a simple technique. This study also provided an interesting insight to the further understanding of the variation of T6SS-5 genes among *B. pseudomallei*, and the uniqueness of certain genes that can only be found in *B. pseudomallei* species.

## Figures and Tables

**Figure 1 diagnostics-12-00562-f001:**
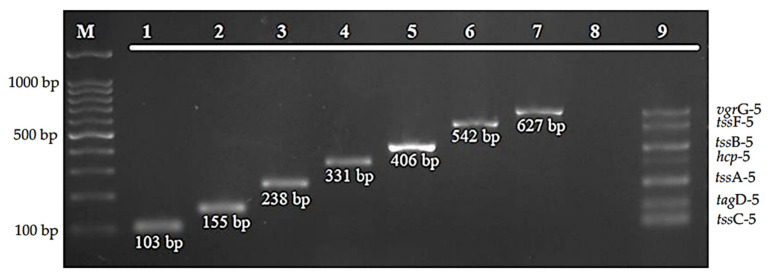
Amplification of the target genes by the developed multiplex PCR assay. Lane M: 100 bp DNA marker; lanes 1–7: monoplex PCR of *tss*C-5, *tag*D-5, *tss*A-5, *hcp*-5, *tss*B-5, *tss*F-5, *vgr*G-5, respectively; lane 8: negative control; and lane 9: multiple bands of all seven genes.

**Figure 2 diagnostics-12-00562-f002:**
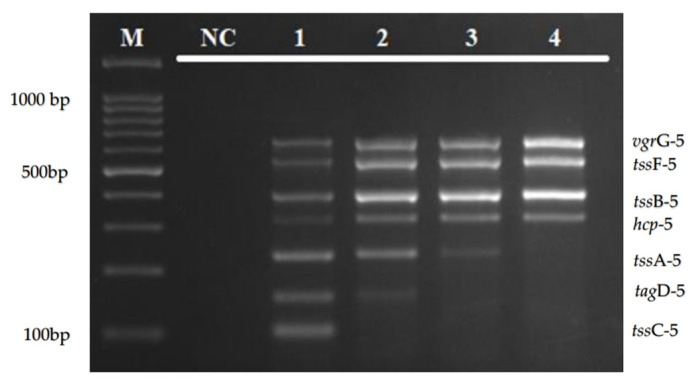
Optimization of primer concentration for *tss*A-5, *tss*C-5, and *tag*D-5 primers. Lane M: 100 bp DNA marker; lane NC: negative control; lane 1: 5.0 µm; lane 2: 2.5 µm; lane 3: 1.25 µm; and lane 4: 0.625 µm.

**Figure 3 diagnostics-12-00562-f003:**
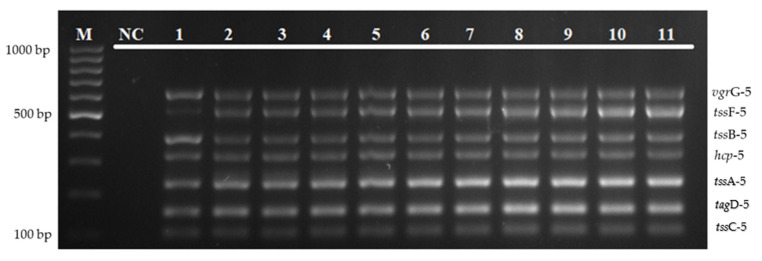
Agarose gel electrophoresis for the optimization of annealing temperature using gradient PCR. The optimization of annealing temperature for the multiplex PCR assay was set up from 52 °C to 62 °C. Lane M: 100 bp DNA ladder; lane NC: negative control; and lanes 1–11: annealing temperature from 52 °C to 62 °C, respectively.

**Figure 4 diagnostics-12-00562-f004:**
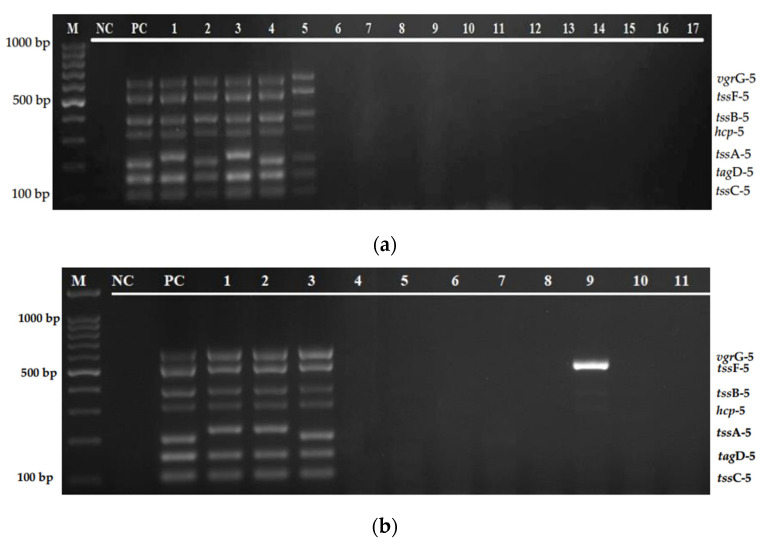
(**a**) Specificity evaluation of the multiplex PCR assay on *B. pseudomallei* isolates and other bacterial strains. Lane M: 100 bp DNA ladder; lane NC: negative control; lane PC: positive control; lanes 1–5: *B. pseudomallei* isolates; and lanes 6–17: *S. aureus* ATCC 25923, *K. pneumoniae* ATCC 70063, *E. faecalis* ATCC 29212, *P. aeruginosa* ATCC 27853, *V. parahaemolyticus* ATCC 17802, *N. meningitidis* 13090, *E. coli* ATCC 35218, *Streptococcus* Group A, *S. epidermidis*, *S. enterica serovar Thypimurium*, *Streptococcus* Group B, and *S. viridans*, respectively. (**b**) Specificity evaluation of the multiplex PCR assay on *B. pseudomallei* isolates and other bacterial strains. Lane M: 100 bp DNA ladder; lane NC: negative control; lane PC: positive control; lanes 1–3: *B. pseudomallei* isolates; lanes 4–6: *S. maltophilia* isolates; lanes 7–11: *H. pylori*, *E. faecalis*, *V. cholerae*, *Salmonella* sp., and *P. mirabilis*, respectively.

**Figure 5 diagnostics-12-00562-f005:**
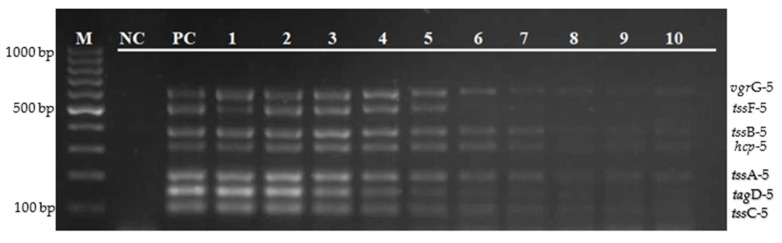
Analytical sensitivity of multiplex PCR assay in the presence of 10^9^ to 10^0^ CFU/mL of *B. pseudomallei* concentrations. Lane M: 100 bp DNA ladder; lane NC: negative control; lane PC: positive control; and lanes 1–10 represent 10^9^ to 10^0^ CFU/mL lysate of *B. pseudomallei*, respectively.

**Figure 6 diagnostics-12-00562-f006:**
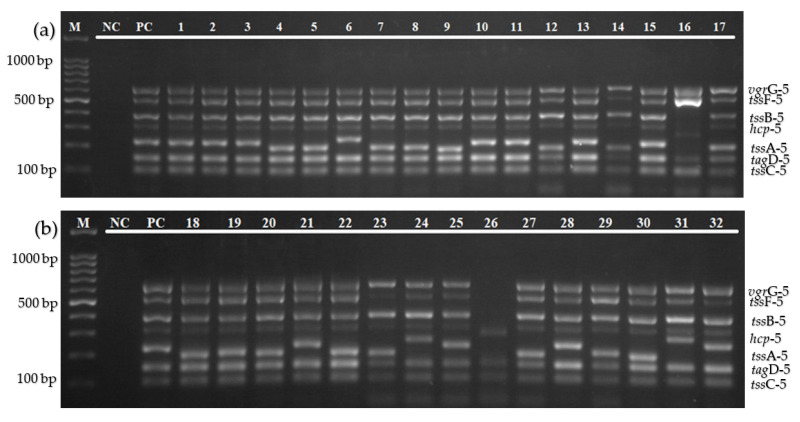
(**a**,**b**): Evaluation of multiplex PCR using clinical samples. Lane M: 100 bp DNA ladder; NC: negative control; PC: positive control; and lanes 1–32 represent clinical isolates of *B. pseudomallei*.

**Figure 7 diagnostics-12-00562-f007:**
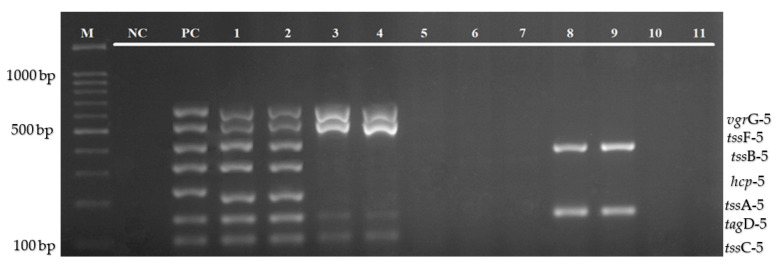
Evaluation of multiplex PCR using environmental isolates of *B. pseudomallei* and other species of Bukrholderia. Lane M: 100bp DNA Ladder; NC: negative control; PC: positive control; lanes 1–2: *B. pseudomallei*; lanes 3–4: *B. thailandesis*; lane 5: *B. ubonensis*; and lanes 6–11: *B. stagnalis*.

**Table 1 diagnostics-12-00562-t001:** Primers used for the optimization of the multiplex PCR assay.

No.	Primer Pairs	Primer Sequences (5′ to 3′)	Target Gene	Amplicon Size (bp)
1	tssC_F	GAGCTTCGCAGACTATCGCT	*tss*C-5	103
	tssC_R	GATCTCGCCCATCGATTCGT		
2	tagD_F	ATGTCGGCGAAGATGATGGG	*tag*D-5	155
	tagD_R	ATCACTTTCTGCTGGCTCGG		
3	tssA_F	GCCGGATCAATCAAAGCCTG	*tss*A-5	238
	tssA_R	TTGAGGTGGTTGAGGTGGTG		
4	hcp_F	CCAGGGGGAAATCAAAGGCT	*hcp-5*	331
	hcp_R	GGGCGAGTATTGGTCCATGT		
5	tssB_F	GATCCGTCGCACCCAAAGAG	*tss*B-5	406
	tssB_R	CTGCGAAAGCCGGGAATGTT		
6	tssF_F	GAACCTGCTGTTTCCGCACT	*tss*F-5	542
	tssF_R	CCAGCTCGACGAACAGGAAT		
7	vgrG_F	CACCTGCTGTTTCCCGATCT	*vgr*G-5	644
	vgrG_R	ATCGACACCGAGCACTTGAG		

## Data Availability

Not applicable.

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
