# Peer review of "Development of Multiplex PCR Assay for Screening of T6SS-5 Gene Cluster: The Burkholderia pseudomallei Virulence Factor"

_diagnostics, 2022, doi:10.3390/diagnostics12030562_

Round 1
Reviewer 1 Report
I read with much interest your work. It is a well-written manuscript and well-designed study showing how Burkholderia pseudomallei, can be identified and distinguished from other bacterial strains by a multiplex PCR. This technique, easy and time-saving, can be very useful for all in places where other techniques which have been developed to identify this bacterium, are too expensive (i.e. realTime PCR doi: 10.1128/JCM.42.12.5871-5874.2004,;https://doi.org/10.3389/fmicb.2020.00072, NG sequencing doi: 10.1371/journal.pgen.1003795, microarray technics and others).
It is shown that this method can distinguish B. pseudomallei from B. thailandesis, B. ubonensis and B. stagnalis and other not Burkholderia strains but there are other strains to be analyzed.
I have only this important point to indicate:
To really demonstrate the specificity of the methodology, authors should try this multiplex amplification, in addition to the strains they show, in other strains, like Burkholderia vietnamiensis, Stenotrophomonas maltophilia, and in particular Chromobacterium violaceum, which can cause an infection with similar symptoms to melioidosis.
Author Response
Response to Reviewer 1 Comments
Point 1:
I read with much interest your work. It is a well-written manuscript and well-designed study showing how Burkholderia pseudomallei, can be identified and distinguished from other bacterial strains by a multiplex PCR. This technique, easy and time-saving, can be very useful for all in places where other techniques which have been developed to identify this bacterium, are too expensive (i.e. realTime PCR doi: 10.1128/JCM.42.12.5871-5874.2004,;https://doi.org/10.3389/fmicb.2020.00072, NG sequencing doi: 10.1371/journal.pgen.1003795, microarray technics and others).
It is shown that this method can distinguish B. pseudomallei from B. thailandesis, B. ubonensis and B. stagnalis and other non Burkholderia strains but there are other strains to be analyzed.
Response 1:
First of all, thank you for taking the time to assess our manuscript. Thank you again for the positive comments. We have included an additional statement based on your comments in the conclusion part (page 10, lines 317-324).
Point 2:
To really demonstrate the specificity of the methodology, authors should try this multiplex amplification, in addition to the strains they show, in other strains, like Burkholderia vietnamiensis, Stenotrophomonas maltophilia, and in particular Chromobacterium violaceum, which can cause an infection with similar symptoms to melioidosis.
Response 2:
We really appreciate your fruitful suggestion. Therefore, we have added a list of non-Burkholderia strains in the methodology section (page 4, lines 141-142), which include Vibrio cholerae, Helicobacter pylorii, Proteus mirabilis, and the suggested Stenotrophomonas maltophilia, to convince the specificity of this assay as recommended by reviewer 1. However, we do not have Burkholderia vietnamiensis and Chromobacterium violaceum in our lab. Details of the result were added (page 6, lines 192-195) and described on page 9 (lines 274-277 and lines 291-292). The abstract part was also edited accordingly (page 1, lines 20 and 24). In total, 20 isolated DNA from non-Burkholderia species were used for the specificity test of this assay.
Reviewer 2 Report
Excellent work. I would insist that results are dependent on the quality/quantity of DNA input - i.e. they must be done from the colony...
Rapid and reliable multiplex PCR assay for the detection of T6SS-5 virulence gene cluster of B. pseudomallei, but ONLY FROM PURE COLONY. That must be clear since possible misinterpretation of direct detection from "row" clinical samples should not be made.
Author Response
Response to reviewer 2
Point 1
Excellent work. I would insist that results are dependent on the quality/quantity of DNA input - i.e. they must be done from the colony...
Rapid and reliable multiplex PCR assay for the detection of T6SS-5 virulence gene cluster of B. pseudomallei, but ONLY FROM PURE COLONY. That must be clear since possible misinterpretation of direct detection from "row" clinical samples should not be made.
Response 1
Thank you very much for your positive comments. It is true that in this study, we tested the assay merely on pure colonies of clinical and environmental isolates as an initial laboratory evaluation. We have made it clear regarding the pure colonies on page 9 (line 237-238) and (line 316) in the discussion and conclusion part, respectively. We have added a future recommendation to evaluate this assay directly from clinical specimens for further diagnostic evaluation (page 9, lines 304-306). Thank you.
Round 2
Reviewer 1 Report
Thanks for your answers.